# Spatial–Temporal Evolution and Improvement Measures of Embodied Carbon Emissions in Interprovincial Trade for Coal Energy Supply Bases: Case Study of Anhui, China

**DOI:** 10.3390/ijerph192417033

**Published:** 2022-12-18

**Authors:** Menghan Zhang, Suocheng Dong, Fujia Li, Shuangjie Xu, Kexin Guo, Qian Liu

**Affiliations:** 1Research Office of Resource Economics and Energy and Mineral Resources, Institute of Geographic Sciences and Natural Resources Research, CAS, Beijing 100101, China; 2College of Resources and Environment, University of Chinese Academy of Sciences, Beijing 100049, China; 3Innovation Academy for Green Manufacture, Chinese Academy of Sciences, Beijing 100190, China; 4College of Geographical Sciences, Liaoning Normal University, Dalian 116029, China

**Keywords:** regional economic integration (REI), embodied carbon emissions in interprovincial trade (ECEs-IPT), energy supply bases (ESBs), spatial–temporal evolution, improvement measures, Anhui

## Abstract

On account of the long-term dependence on energy trade and the phenomenon of embodied carbon emissions in interprovincial trade (ECEs-IPT), energy supply bases (ESBs) in the economic integration regions (EIRs) are under unprecedented dual pressure of achieving carbon emissions (CEs) reduction targets and ensuring security and stability of the energy supply. This problem has attracted more and more attention and research by experts and scholars. This paper took Anhui, the coal ESB of the Yangtze River Economic Belt (YREB), as an example and took the key stage of rapid development of regional economic integration (REI) and accelerated the realization of CEs reduction targets in YREB from 2007 to 2017 as the study period. From the perspectives of regions and industry sectors, we calculated the transfer amount of ECEs-IPT in Anhui among the YREB, analyzed the spatial–temporal evolution pattern of ECEs-IPT, and revealed the industrial characteristics of ECEs-IPT. Then, we classified the industry sectors and proposed the direction of industrial improvement measures. The results showed that, during the decade, the amount of provinces undertaking the net ECEs-IPT outflow from Anhui increased significantly and spatially expanded from only Jiangxi Province to almost all of the YREB. In addition, 39.77% of the net ECEs-IPT outflow of Anhui was concentrated in petroleum processing, coking, and nuclear fuel processing (RefPetraol), metal smelting and rolling processing (MetalSmelt), and electricity and heat production and supply (ElectpowerProd) that trade with Shanghai, Jiangsu, Zhejiang, and Jiangxi. The analytical model and results will provide a useful reference for the global similar coal ESBs, especially the coal ESBs within the EIRs, to formulate improvement measures for regions or even the world to ensure stability of the energy supply and achieve regional CEs reduction targets.

## 1. Introduction

Nowadays, with the in-depth combined impacts of the global economic crisis, climate change, and energy crisis, controlling carbon emissions (CEs) (Appendix A) has become the largest challenge currently faced by all countries in the world [1,2,3,4,5]. To accelerate the achievements of global CEs reduction targets, some carbon-intensive economic integration regions (EIRs) with fast-growing economies, such as the EU, SAARC, and BRICS, are using regional economic integration (REI) to develop regional CEs reduction agreements to jointly address climate change challenges and potential environmental threats [6,7]. Compared to general inter-country and inter-regional trade activities, economic activities within EIRs are relatively more independent, have stronger economic ties, have fewer barriers to trade, and trade commodities more frequently [8,9]. However, differences in economic development levels, resource advantages, and industry sector types within EIRs can further exacerbate the risks of embodied CEs in interprovincial trade (ECEs-IPT) behind commodity flows [10,11,12].

In recent years, transfer of ECEs-IPT in EIRs has attracted widespread attention from experts and scholars and has become one of the research hotspots in the field of CEs embodied in interregional trade [13,14,15,16]. Cai et al. (2018) calculated the embodied CEs from imports and exports of countries along the Belt and Road and showed that China, as an important energy supply area, has been responsible for embodied CEs transfer from the trade of 22 developed countries [17]. Tian et al. (2022) found that complete elimination of tariffs by 13 members of the Regional Trading Partners would increase global CEs from fuel combustion by about 3.1% per year and lead to a rapid increase in CEs from developing countries [7]. As the degree of mutual openness within internal regions continues to increase, economic ties continue to strengthen, and REI has become increasingly prominent. However, current academic research has mainly focused on transfer of ECEs-IPT in EIRs between countries or regions, and transfer of ECEs-IPT in EIRs established within countries has not been sufficiently studied. Although some scholars have studied the scale and characteristics of ECEs-IPT in a particular economic zone within a country, most of them have remained only in static analysis of a certain year, analysis of transfer direction, and measurement of the transfer amount [18,19,20,21]. Moreover, there are few studies on the dynamic spatial–temporal evolution of embodied carbon transfer processes and driving mechanisms at long time scales for regions with a high degree of REI and relatively obvious intra-regional variability in economic levels. Therefore, studies on identification of the potential risks of embodied carbon emissions transfer in the process of REI need to be supplemented and improved. Meanwhile, this lack of research also hinders further development of in-depth research on improvement measures for EIRs within countries.

China is in a phase of rapid REI [22,23]. The Chinese government has been vigorously promoting development of REI [24]. Many EIRs in China, such as the Beijing–Tianjin–Hebei metropolitan area, the Yangtze River Economic Belt (YREB), and the Guangdong–Hong Kong–Macao Greater Bay Area, have already generated significant economic achievements [25,26,27,28,29,30]. Among them, the YREB performs the best [31,32], but there are also potential risks of ECEs-IPT that need to be identified and regulated. The YREB is relatively scarce in coal resources [33], with only Guizhou, Anhui, Yunnan, and Chongqing being rich [34,35,36]. In the process of REI, they provided long-term energy security for energy-resource-poor regions and cities among the YREB through coal-fired power generation and hydropower generation. The economic benefits within the region are maximized through complementary advantageous resources. Due to its geographical proximity to key cities, such as Shanghai, Nanjing, Suzhou, and Wuhan, Anhui has gradually become one of the most important coal energy supply bases (ESBs) in the YREB [37]. Anhui has been responsible for ensuring the energy security of the YREB, meeting the basic energy needs, and supporting the smooth and orderly development of the regional economy and society. The existence of ECEs-IPT has caused Anhui to bear a large amount of carbon leakage from other provinces, which seriously affects the achievement of its CEs reduction targets [38,39,40]. Therefore, in the context of REI, there is an urgent need to study the role of regional synergistic development in CEs reduction and to explore the potential of EIRs in reducing CEs. On this basis, it is of crucial practical significance to strengthen the research on the evolution pattern, mechanism, and regulation of ECEs-IPT in coal ESBs in order to relieve the dual pressure of achieving CEs reduction targets and ensuring the security and stability of the energy supply.

To reflect more clearly and accurately the characteristics and problems of ECEs-IPT in coal ESBs, we select the YREB in China as the study area. The YREB is a typical representative of high REI with a high degree of economic system independence, high level of economic development, high CEs, and high cross-regional trade. Meanwhile, Anhui, a coal ESB within the YREB, is selected as the study case, which is still at the stage of high CEs. This paper built a multiregional input–output (MRIO) model based on the input–output tables of 30 Chinese provinces and regions in 2007 and 2017. From the perspectives of regions and industry sectors, we calculated the transfer amount of ECEs-IPT in Anhui among the YREB, analyzed the spatial–temporal evolution pattern of the ECEs-IPT, and revealed the industrial characteristics of ECEs-IPT. On this basis, a coupling relationship model between economic and environmental effects was established to classify the types of industry sectors and to propose improvement measures based on the industrial regulation directions. The analytical model and the results will directly provide scientific support for Anhui in solving the difficulty of ensuring energy security and achieving CEs reduction goals in the context of REI. Meanwhile, it will provide a useful reference for the global similar coal ESBs, especially the coal ESBs within the EIRs, to formulate improvement measures for regions or even the world to ensure stability of the energy supply and achieve regional CEs reduction targets.

## 2. Study Area

China has the largest CEs volume and total energy consumption in the world [41]. Fossil energy has dominated the energy consumption system of China for a long time. Among the rest, coal is the most important [42], which accounted for 56.0% of primary energy consumption in 2021 [43]. Anhui (Figure 1), the case study area, is one of the few coal-rich provinces and coal ESBs in southern China. It has the highest average production capacity of coal mines in China, with more than 30% of the coal reserves in the YREB, ranking second. The YREB has an economically high level but a relative lack of coal resources, with only Guizhou, Anhui, Yunnan, and Chongqing being rich in coal resources.

With the release of the *Outline of the Yangtze River Delta* (*YRD*) *Regional Integrated Development Plan* [44], the whole of Anhui has been included in the integrated development of the YRD. This marked further elevation in the strategic position of Anhui in the YREB and even in the whole country development pattern. *The CPC Anhui Provincial Committee’s Proposal on Formulating the 14th Five-Year Plan for National Economic and Social Development and Visionary Targets for 2035* pointed out that Anhui could implement the energy supply guarantee project, support the construction of coal reserve bases and other projects in the YRD, construct intelligent energy systems, and establish a clean, low-carbon, safe, and efficient modern energy system [45]. Compared to the western and northern provinces, Anhui has obvious location advantages, which is closer to more economically developed regions in YREB, such as YRD and the middle reaches of the Yangtze River urban agglomeration. In 2021, 112.55 Mt raw coal were produced in Anhui, of which approximately 85% were consumed locally and 15% were exported to Jiangsu, Shanghai, Zhejiang, Hubei, and Hunan. Coal (including electricity) exported outside accounted for approximately 36% of coal production. As an important node province in the YREB, Anhui has an indispensable and important role in the process of energy supply and high-quality development.

However, production and trade of energy-intensive energy products have also brought Anhui plenty of ECEs-IPT, making the CEs grow rapidly. In particular, between 2007 and 2019, the CEs of Anhui grew from 395.31 Mt to 816.13 Mt, with an average annual growth rate of 6.23%, making it the province with the fastest growth in CEs and the second highest CEs in the YREB (Figure 2). Meanwhile, Anhui was also the province with the largest increase in net ECEs-IPT in the YREB during the decade from 2007 to 2017 (Figure 3). In addition, Anhui has changed from a region of net ECEs-IPT inflow in 2007 to a region of net ECEs-IPT outflow in 2017 and began to bear carbon leakage from upstream and downstream regions in YREB with a deepening degree. As a result, Anhui is facing urgent practical pressure to ensure energy demand and achieve CEs reduction targets.

The situation of Anhui reflects the typical characteristics of the regional coal ESBs suffering from the dual pressure of achieving CEs reduction targets and ensuring security and stability of the energy supply. Therefore, this study selected Anhui as a study area to analyze the spatial–temporal evolution processes, the mechanism of the ECEs-IPT in Anhui among the YREB, and proposed targeted improvement measures. This paper provided a scientific reference for similar regional coal ESBs around the world to scientifically strengthen the stability of the energy supply while achieving CEs reduction targets in the process of REI.

## 3. Materials and Methods

This research was based on the China regional expansion input–output (IO) table in 2007 and 2017, used MRIO model (Section 3.1) and ECEs-IPT estimation model (Section 3.2), calculated the amount of ECEs-IPT in Anhui among the YREB, analyzed the spatial–temporal evolution pattern of ECEs-IPT, and revealed the industrial characteristics of ECEs-IPT. On this basis, we established a coupling relationship (CR) model (Section 3.3) to classify the industry sectors and proposed the improvement measures for different industrial types (Figure 4).

### 3.1. MRIO Model

The IO analysis method was developed by economist Leontief in the mid-1930s [46], widely used and extended to the environment [47], and then to CEs research [48]. IO analysis is the mainstream method for ECEs-IPT at the national and regional levels. Accounting of input value and output value can not only reflect the direct economic and technological connections, which exist in the production process of each production department, but also reveal other economic and technological connections hidden between each production department and easily neglected. The MRIO table is the basis of IO analysis, which can fully present the source of the input cost in the production process and intermediate use of various products and services in the output process. Its basic structure is shown in Table 1.

The MRIO table consists of the original IO table (the matrix of the upper left) and the expansion IO tables. The expansion IO tables include the “interprovincial outflow” table (the matrix of the upper right) and the “interprovincial inflow” table (the matrix of the lower left). Each number $x_{r}^{ij}$ can be interpreted from the vertical and horizontal directions: Horizontally, $x_{r}^{ij}$ (*i*,*j* = 1…*n*, *r* = 1…*m* (in our research, *n* = 28, *m* = 30) represents the intermediate use in region r produced by industry sector *i* to industry sector *j* to use; vertically, which also represents intermediate input in the process of production. The original IO table consists of the final use, the total output, the added value, and the total input. Among them, $F_{r}^{i}$ refers to the final use of industry sector *i*, $X_{r}^{i}$ refers to the total output of industry sector *i*, $VA_{r}^{j}$ refers to the added value of industry sector *j*, and $Y_{r}^{j}$ refers to the total input of industry sector *j* (*r* = 1⋯*m*; in our research, *m* = 30). $PEX_{rd}^{i}$ in “interprovincial outflow” table represents the commodity value (CV) transfer from region r to region d of industry sector *i*, and $PIM_{rd}^{j}$ in “interprovincial inflow” table represents the CV transfer from region d to region *r* of industry sector *j* (*d*,*r* = 1⋯*m*; in our research, *m* = 30).

The matrix of the upper left and the matrix of the upper right are joined together to form a horizontal table, which can reflect the use of products and services produced by various industry sectors. The vertical table, formed by the matrix of the upper left and the matrix of the lower left, reflects the various sources of input and the value composition of products in the production. That is why the MRIO table can reflect the value formation process of products and services of all industry sectors in the national economy system. According to the MRIO table, we obtained:(1)$$X={\left(I-A\right)}^{-1}Y\;$$
where *A* is the matrix of local technical coefficient matrix, ${\left(I-A\right)}^{-1}$ is generally called the Leontief inverse matrix or the cumulative coefficient matrix, *X* represents the matrix of total output matrix, *Y* is the matrix of final use for all regions.

Here, if using the value outflow in interregional trade $E_{x}$ instead of *Y*, we can calculate the total input for production of $E_{x}$ as Equation (2):(2)$$X_{e}={\left(I-A\right)}^{-1}E_{x}\;$$

### 3.2. ECEs-IPT Estimation Model

To measure ECEs-IPT, the energy consumption coefficient and direct CEs coefficient should be introduced successively based on the MRIO model to calculate the amount of embodied energy and embodied CEs, respectively.

In order to calculate the amount of embodied energy, it is necessary to calculate the energy consumption coefficient of each region and each industry sector according to the existing total energy consumption data, that is, the energy consumed per unit of output value.
(3)$$e_{r}=C_{r}/X_{r}\;$$
where *C_r_* represents the total energy consumption of region *r*, $X_{r}$ represents the total output of region *r*, $e_{r}$ expresses the energy consumption coefficient of region *r*, which reflects conversion from different types of energy consumption to standard coal.

The product of the energy consumption coefficient $e_{r}$ and output value $X_{e}$ is the total energy consumption included in the adjustment, also known as embodied energy, represented as *E*, and the following formula can be obtained:(4)$$E=e\cdot \left[\left(I-A\;\right){\;}^{-1}\;E_{x}\right].$$

According to CEs coefficient μ, convert standard coal into embodied CEs and calculate the amount of ECEs-IPT as follows:(5)$$C_{e}=\mu \times E=\mu \times e_{r}\times \left[{\left(I-A\right)}^{-1}E_{x}\right]\;$$

### 3.3. Coupling Relationship (CR) Model

Due to the complex interaction of various elements in MRIO analysis, reducing the output value of any industry sector will inevitably lead to a decrease in value volume in upstream and downstream industry sectors, resulting in a multiplier effect and huge economic losses. Taking economic and environmental effects into consideration, we could find out the key sectors and minimize the environmental damage caused by transfer of ECEs-IPT under the premise of maximizing the economic benefits. Here, we referred to the method proposed by Cheng et al. [8] to establish a coupling relationship (CR) model. The CR model has advantages in quantitative evaluation of the coupling relationship between economic and ecological environmental benefits. This model can help us analyze the coupling relationship between the change in net CV outflow and net ECEs-IPT outflow, which is in line with our research need about simultaneously analyzing the economic benefits and the environmental negative benefits, such as transfer of ECEs-IPT.

According to the CR model, we used contribution ratio to calculate the change in net CV outflow ($\Delta nR_{E}$) and net ECEs-IPT outflow ($\Delta nR_{c}$):(6)$$\left\{\begin{array}{c} \Delta nR_{E}=nP_{E}^{1}-nP_{E}^{0}=\frac{nE_{x}^{1}}{E_{total}^{1}}-\frac{nE_{x}^{0}}{E_{total}^{0}}\\ \Delta nR_{c}=nP_{C}^{1}-nP_{C}^{0}=\frac{nC_{x}^{1}}{C_{total}^{1}}-\frac{nC_{x}^{0}}{C_{total}^{0}} \end{array}\right.\;$$
where $\Delta nR_{E}$ and $\Delta nR_{c}$ represent the changes in the contribution ratio of net CV outflow and net ECEs-IPT outflow, $E_{total}^{y}$ (*y* = 0, 1) represents the total CV, $C_{total}^{y}$ (*y* = 0, 1) represents the total ECEs-IPT, both of them including outflow and inflow. The $nE_{x}^{y}$ and $nC_{x}^{y}$ represent that the net CV outflow and net ECEs-IPT outflow of industry sector *x* (*x*= 1…*n*, *n* = 28 in our research) in year *y* (*y* = 0, 1) represent 2007 and 2017, respectively, in our research).

After calculating the $\Delta nR_{E}$ and ($\Delta nR_{c}$ of each industry sector; we divided sectors into four types, with five subtypes according to the coupling relationships between them, as follows: Type A ($\Delta nR_{E}$ > 0 and $\Delta nR_{c}$ < 0); Type B ($\Delta nR_{E}$ < 0 and $\Delta nR_{c}$ > 0); Type C ($\Delta nR_{E}$ > 0 and $\Delta nR_{c}$ > 0), includes Type C-I ($\Delta nR_{E}$ > $\Delta nR_{c}$ > 0) and Type C-II ($\Delta nR_{c}$ > $\Delta nR_{E}$ > 0); Type D ($\Delta nR_{E}$ < 0 and $\Delta nR_{c}$ < 0). Based on the above analysis, we proposed improvement measures for the different sectors characterized by different types: Type A should implement the promoted measures; Type B and Type D should implement the controlled measures; Type C-I and Type C-II should implement the orientation-encouraged and orientation-reduced measures, respectively.

### 3.4. Data Sources

The interprovincial trade data come from the MRIO tables in 2007 (excluding Tibet) and in 2017 (including Tibet) for China, compiled by the Development Research Center of the State Council, P.R.C. [49,50]. Both exclude Hong Kong, Macao, and Taiwan region.

The data about energy consumption were collected from the “Energy Consumption by Industry Sector” table from Statistical Yearbooks of 30 Provinces in 2007 and 2017. If there is none in this table, we replaced it with the “Energy Balance of Region” table in the China Energy Statistical Yearbooks. Based on energy consumption data, we estimated the coefficients of energy consumption for 28 industry sectors in 30 provinces with the guidance of the IPCC reference approach [51,52].

Finally, we converted the actual value of energy consumption data into standard coal of that by energy conversion coefficient, and as well converted into CEs data by the CEs coefficient. The energy conversion coefficient and CEs coefficient provided by the IPCC and Energy Statistics Knowledge Manual [53].

## 4. Results

### 4.1. Changes in the Net ECEs-IPT Outflow in Anhui among the YREB

In the decade between 2007 and 2017, the net ECEs-IPT outflow in Anhui among the YREB showed a significant fast-increasing tendency, from −9.25 Mt in 2007 to 23.04 Mt in 2017, which illustrated a shift from a region of net ECEs-IPT inflow in 2007 to a region of net ECEs-IPT outflow in 2017. This indicated that, along with interprovincial commodity trade, Anhui has changed its role in synergistic development of the YREB and gradually started to take on large amounts of carbon leakage from more developed provinces in the eastern coast and southwest China to support economic development of these provinces.

The net ECEs-IPT outflow in Anhui among the YREB as a rate of the total CEs in Anhui increased from 17.58% in 2007 to 27.08% in 2017 (Figure 5a), with an average annual growth rate of 9.56%, while the net CV outflow in Anhui among the YREB as a rate of the total regional output in Anhui decreased from 6.60% in 2007 to 3.80% in 2017 (Figure 5b).

The above suggested that Anhui has hidden a large latent ECEs-IPT outflow among the YREB and the scale is increasing. However, the CV outflow is decreasing, reflecting the fact that more embodied CEs rather than CV were transferred in the process of interprovincial commodity trade in Anhui among the YREB. It is quite clear that there is serious inequity in Anhui between economic development and CEs reduction among the YREB. Anhui has been responsible for a significant amount of carbon leakage among the YREB at the cost of its economy. Against the backdrop of the overall success of CEs reduction in the YREB, the growing problem of ECEs-IPT in Anhui will become a key constraint to continued deepening of CEs reduction in the YREB. 

### 4.2. Spatial–Temporal Evolution of the Net ECEs-IPT Outflow in Anhui among the YREB

Quantitatively, the amount of provinces that received net ECEs-IPT outflow from Anhui increased significantly, from one in 2007 to seven in 2017. Spatially, only Jiangxi received net ECEs-IPT outflow in 2007 (Figure 6a). In 2017, the main target provinces for net ECEs-IPT outflow gradually extended to almost all the YREB except for Jiangsu and Shanghai in the downstream regions of the YREB (Figure 6b). Moreover, in 2017, the major target provincial regions of net ECEs-IPT outflow were Zhejiang, Jiangxi, and Chongqing, with 14.31 Mt, 4.12 Mt, and 2.01 Mt, respectively, and the total contribution ratio was 89% for Anhui.

Meanwhile, the amount of provinces that received net CV outflow from Anhui also increased rapidly, from two in 2007 to eight in 2017. Spatially, Zhejiang, Chongqing, and Jiangxi received net CV outflow in 2007, with CNY 21.13 billion, CNY 6.65 billion, and CNY 6.08 billion, respectively. Meanwhile the total contribution ratio of Anhui reached 91%. Thus, Anhui provided a considerable CV to the YREB, especially Zhejiang, Chongqing, and Jiangxi. Although interprovincial trade has met the production and consumption needs of these developed provinces, significant carbon leakage occurred.

### 4.3. Industry Sectors Characteristics of Net ECEs-IPT Outflow in Anhui among the YREB

In terms of industry sectors, in 2007, the top five regions and industry sectors in the net ECEs-IPT outflow in Anhui among the YREB (Figure 7a) were Coalmin (see Appendix A for specific information of 28 industry sectors) from Anhui to Zhejiang (S02, −2.16 Mt), ElectpowerProd from Anhui to Jiangsu (S22, −1.70 Mt), Transport from Anhui to Shanghai (S26, −0.99 Mt), MetalSmelt from Anhui to Jiangsu (S14, −0.90 Mt), and RefPetraol from Anhui to Jiangsu (S11, 0.85 Mt). 

In 2017, the top five regions and industry sectors in the net ECEs-IPT outflow in Anhui among the YREB (Figure 7b) were ElectpowerProd from Anhui to Zhejiang (S22, 5.84 Mt), NonMProd from Anhui to Zhejiang (S13, 5.83 Mt), MetalSmelt from Anhui to Shanghai (S14, −2.51 Mt), RefPetraol from Anhui to Jiangxi (S11, 2.49 Mt), and Textile from Anhui to Zhejiang (S07, −1.52 Mt). 

During the decade, three industry sectors in Anhui, RefPetraol, MetalSmelt, and ElectpowerProd, maintained their dominant position in interprovincial trade with Shanghai, Jiangsu, Zhejiang, and Jiangxi four provinces, and the total contribution of the above regions and industry sectors to the net ECEs-IPT outflow increased from 28.74% in 2007 to 39.77% in 2017.

## 5. Discussion

### 5.1. Improvement Measures for ECEs-IPT Reduction of Different Industry Sectors

#### 5.1.1. Classification for Industry Sector by CR Model

According to the changing evolution trend for the coupling relationship between the change in net CV outflow and net ECEs-IPT outflow of each industry in Anhui among the Yangtze River Economic Belt from 2007 to 2017 (Figure 8), the 28 industry sectors were divided into four types, with five subtypes, as follows: Type A ($\Delta nR_{E}$ > 0 and $\Delta nR_{c}$ < 0); Type B ($\Delta nR_{E}$ < 0 and $\Delta nR_{c}$ > 0); Type C ($\Delta nR_{E}$ > 0 and $\Delta nR_{c}$ > 0), includes Type C-I ($\Delta nR_{E}$ > $\Delta nR_{c}$ > 0) and Type C-II ($\Delta nR_{c}$ > $\Delta nR_{E}$ > 0); Type D ($\Delta nR_{E}$ < 0 and $\Delta nR_{c}$ < 0). 

#### 5.1.2. Improvement Measures for Different Industrial Types

Based on the classification of industry sector types, different improvement measures for each industrial type were proposed (Figure 9). For industry sectors belonging to Type A, they should implement the promoted policies and the specific improvement measures, such as MetalOreMin (S04) and eight other industry sectors. For industry sectors belonging to Type B and Type D, they should implement the controlled policies and the specific improvement measures, such as Coalmin (S02) and ElectpowerProd (S22) in Type B and Agri (S01) and twelve other industry sectors in Type D. For industry sectors belonging to Type C-I, they should implement orientation-encouraged policies and specific improvement measures, such as Textile (S07) and three other industry sectors. For industry sectors belonging to Type C-II, they should implement orientation-reduced policies and specific improvement measures. 

### 5.2. Further Improvement Measures of Different Target Provinces in Major Industry Sectors

For simultaneously relieving the unprecedented pressure of achieving CEs reduction targets and ensuring the security and stability of the energy supply faced by Anhui, it is urgent to reduce CEs by giving full play to the role of regional coordinated development, modifying the industrial structure, and optimizing reallocation of resources. Here, we analyzed the inter-provincial trade structure and scale.

The top three industry sectors of the supreme net ECEs-IPT outflow in Anhui among the YREB in 2017 were ElecMachinery, Coalmin, and NonMProd, with a total contribution of 51%, significantly higher than the other 25 industry sectors (Figure 6). Therefore, we focused on these three industry sectors for further research and proposed further improvement measures from the provincial level.

#### 5.2.1. Improvement Measures for ElecMachinery

According to the results of the industry sector type classification, ElecMachinery (S22) belongs to the controlled type (Type B). In 2017, ElecMachinery was the largest net ECEs-IPT outflow industry sector in Anhui among the YREB, with a net ECEs-IPT outflow of 9.68 Mt and a 21.17% contribution of total.

Considering the structure of interprovincial trade, the contribution rates to net ECEs-IPT outflow from Anhui to Zhejiang, Shanghai, and Jiangxi were relatively larger, with contribution rates of 60.26%, 10.07%, and 7.80%, respectively (Figure 10a), while contribution rates of net CV outflow from Anhui to Chongqing, Sichuan, and Zhejiang were relatively larger, with contribution rates of 34.42%, 26.33%, and 14.32%, respectively (Figure 10d).

Take the trade path between Anhui and Zhejiang as an example: the contribution rate of net CV outflow is 14.32%, while its contribution rate to net ECEs-IPT outflow is 60.26%, which showed seriously unbalanced features. Such a serious inconsistency means that the negative influences caused by the ECEs-IPT with Zhejiang far outweighed the positive influences brought by the CV. The same inconsistency also appeared in the two trade paths from Anhui to Shanghai and Jiangxi. For ElecMachinery, we should implement controlled measures, such as reducing the scale of interprovincial trade with Zhejiang, Shanghai, and Jiangxi. Such control measures would sacrifice a lower economic cost and receive a greater CEs reduction effect.

#### 5.2.2. Improvement Measures for Coalmin

According to the results of the industry sector type classification, Coalmin (S02) belongs to the controlled type (Type B). In 2017, Coalmin was ranked second in terms of the net ECEs-IPT outflow industry sector in Anhui among the YREB, with a net ECEs-IPT outflow of 7.13 Mt and a 15.58% contribution of total.

Considering the structure of the interprovincial trade, the contribution rates to net ECEs-IPT outflow from Anhui to Shanghai, Jiangxi, and Jiangsu were relatively larger, with contribution rates of 20.69%, 20.39%, and 19.71%, respectively (Figure 10b), while the contribution rate of net CV outflow from Anhui to Shanghai was relatively larger, with a contribution rate of 99.60% (Figure 10e).

Take the trade path between Anhui and Shanghai as an example: the contribution rate of net CV outflow is 99.60%, while its contribution rate to net ECEs-IPT outflow is 20.69%, which means that the positive influences brought by the CV with Shanghai far outweighed the negative influences caused by the ECEs-IPT. However, the results of industrial analysis show that the contribution rate of net CV outflow from Anhui to Shanghai has been decreasing in the past decade, while the contribution rate to net ECEs-IPT outflow has been increasing. The long-term high degree of unitary trade object also causes obstacles to relieve the pressure of the transfer of ECEs-IPT in Anhui. Therefore, in the orientation-reduction improvement measures For Coalmin, we should implement controlled measures, such as reducing the scale of interprovincial trade with Shanghai. Such control measures would sacrifice a lower economic cost and receive a greater CEs reduction effect.

#### 5.2.3. Improvement Measures for NonMProd

According to the results of the industry sector type classification, NonMProd (S13) belongs to Promoted type (Type C-I). In 2017, NonMProd was ranked third in terms of the net ECEs-IPT outflow industry sector in Anhui among the YREB, with a net ECEs-IPT outflow of 6.66 Mt and a 14.57% contribution of total.

Considering the structure of interprovincial trade, the contribution rate to net ECEs-IPT outflow from Anhui to Zhejiang was relatively larger, with contribution rates of 87.56% (Figure 10c), while the contribution rate of net CV outflow in the same trade path was relatively larger, with a contribution rate of 90.56% (Figure 10f), which means that the positive influences brought by CV with Zhejiang far outweighed the negative influences caused by ECEs-IPT. Therefore, in the orientation-reduction improvement measures for NonMProd, we should implement orientation-encouraged measures, such as moderately expanding the scale of interprovincial trade with Zhejiang to achieve better CEs reduction effects.

## 6. Conclusions

Placing the problem of ECEs-IPT in coal ESBs in the context of REI can reflect more clearly and intuitively the difficult dilemma faced by coal ESBs to achieve CEs reduction targets and ensure security and stability of the energy supply. Therefore, this study selected Anhui, a coal ESB within the YREB, as a case study, which is still at the stage of high CEs, to analyze the characteristics of ECEs-IPT of coal ESBs in the EIRs, to explore the risks of ECEs-IPT transfer behind interprovincial trade, and to propose several corresponding improvement measures. The main conclusions of this paper are as follows.

(1) Coal ESBs have been a serious problem of embodied carbon leakage, which was highly ignored, especially in the EIRs. This paper has confirmed that Anhui, the coal ESB within the YREB, has been responsible for a large amount of embodied carbon leakage within the YREB at the cost of economic losses from 2007 to 2017. It is worth noting that the embodied carbon leakage from the coal ESBs in the EIRs was easily covered by the good economic development of the whole integrated region. Therefore, this problem has not been studied in depth by academics, and, under the trend of deepening global REI, there is an increasing risk of CEs embodied in interregional trade in the coal ESBs in the EIRs, and continued amplification of this negative impact will cause a decrease in the overall CEs reduction benefits of the whole region. The issue deserves in-depth study.

(2) During the decade from 2007 to 2017, the transfer of ECEs-IPT in Anhui has the characteristics of spatial–temporal evolution, with significant growth in total volume, spatial diffusion, and concentration in high-energy-consuming industry sectors. In terms of total volume, the net ECEs-IPT outflow in Anhui among the YREB has increased from −9.25 Mt to 23.04 Mt, reflecting that Anhui has begun to bear large carbon leakage within the YREB. Spatially, the provinces receiving net ECEs-IPT outflow and net CV outflow from Anhui expanded to almost all provinces of the YREB, indicating that Anhui has taken on carbon leakage from more developed provinces in the eastern coast and southwest China. In terms of industry sectors characteristics, the amounts of carbon leakage undertaken by Anhui were mainly concentrated in three high energy-consuming industry sectors (RefPetraol, MetalSmelt, and ElectpowerProd).

(3) The serious problem of ECEs-IPT in coal ESBs can be regulated by adjusting the inter-provincial trade structure and controlling the inter-provincial trade scale. Based on the analysis of the CR model, this paper proposed the following improvement measures to scientifically adjust and control the structure and scale of inter-provincial trade among provinces and industry sectors within the YREB: control the scale of ElecMachinery from Anhui to Zhejiang, Shanghai and Jiangxi, as well as Coalmin from Anhui to Shanghai, and moderately expand the scale of NonMProd with Zhejiang to achieve better CEs reduction effects. The results will provide a useful reference for similar global coal ESBs, especially the coal ESBs within the EIRs, to formulate improvement measures for regions or even the world to ensure stability of the energy supply and achieve regional CEs reduction targets.

## Figures and Tables

**Figure 1 ijerph-19-17033-f001:**
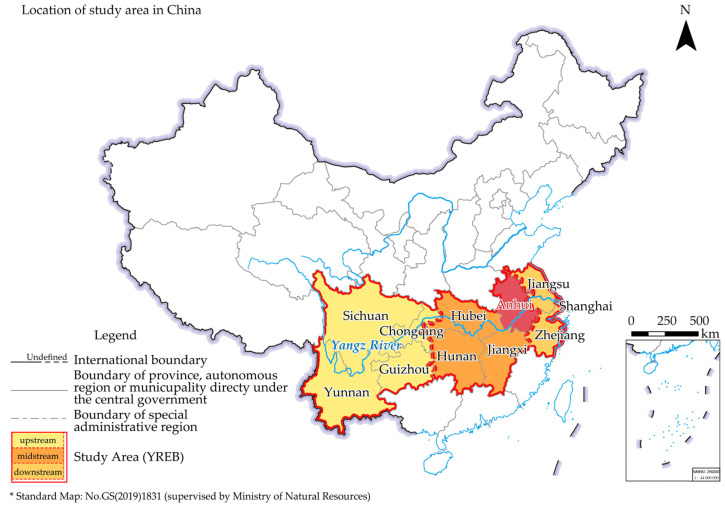
The location of the study area. Attached figure: based on the standard map No.GS (2019)1831 downloaded from the standard map service website of The Ministry of Natural Resources, with no modification.

**Figure 2 ijerph-19-17033-f002:**
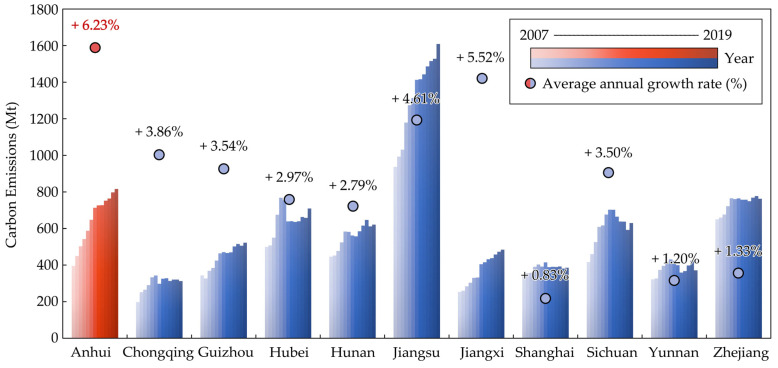
Long time series evolution trend of CEs among the YREB (2007–2019).

**Figure 3 ijerph-19-17033-f003:**
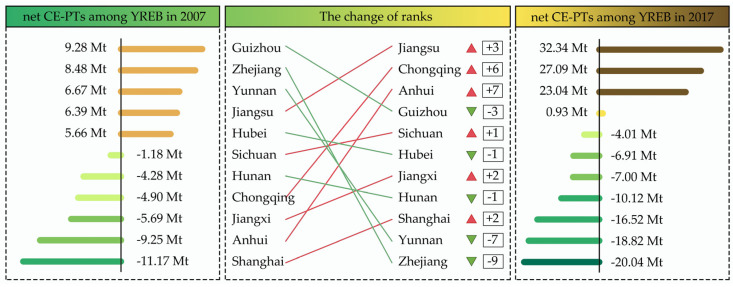
The amount of net ECEs−IPT among the YREB in 2007 and 2017.

**Figure 4 ijerph-19-17033-f004:**
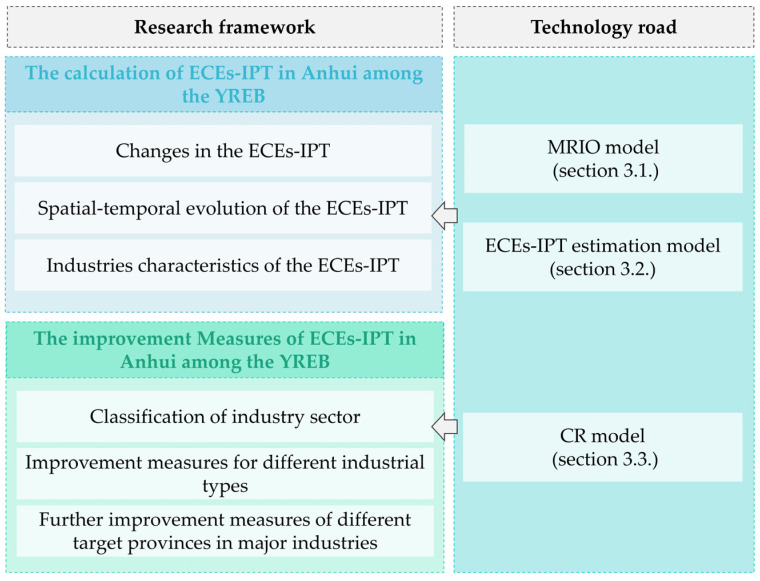
The map of research framework and technology road.

**Figure 5 ijerph-19-17033-f005:**
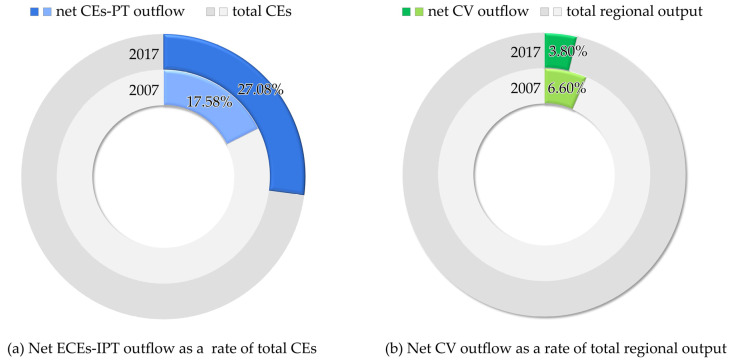
Net ECEs-IPT outflow as a rate of total CEs (**a**) and net CV outflow as a rate of total regional output (**b**) in Anhui among the YREB from 2007 to 2017.

**Figure 6 ijerph-19-17033-f006:**
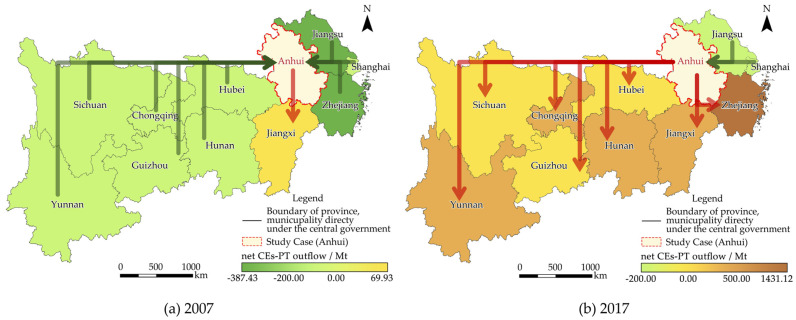
Spatial–temporal evolution of the net ECEs-IPT outflow in Anhui among the YREB from 2007 (**a**) to 2017 (**b**).

**Figure 7 ijerph-19-17033-f007:**
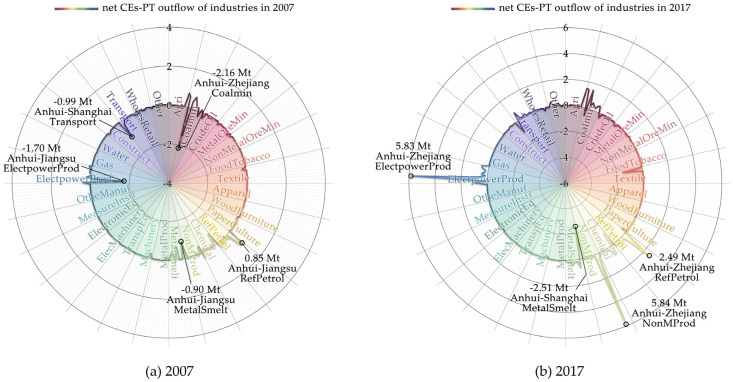
Net ECEs-IPT outflow of industry sectors in Anhui among the YREB from 2007 (**a**) to 2017 (**b**).

**Figure 8 ijerph-19-17033-f008:**
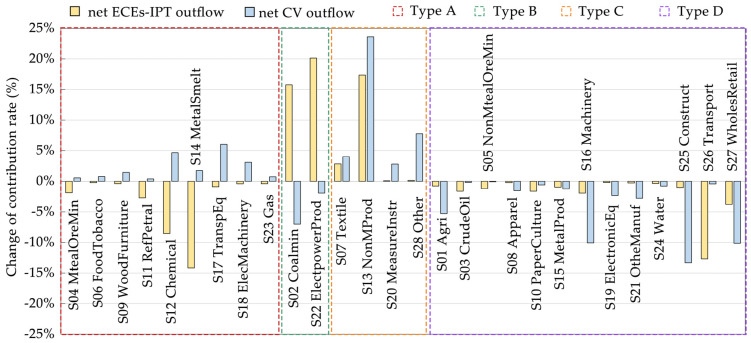
Contribution rate change of net CV outflow and net ECEs−IPT outflow of each industry sector in Anhui among the YREB from 2007 to 2017.

**Figure 9 ijerph-19-17033-f009:**
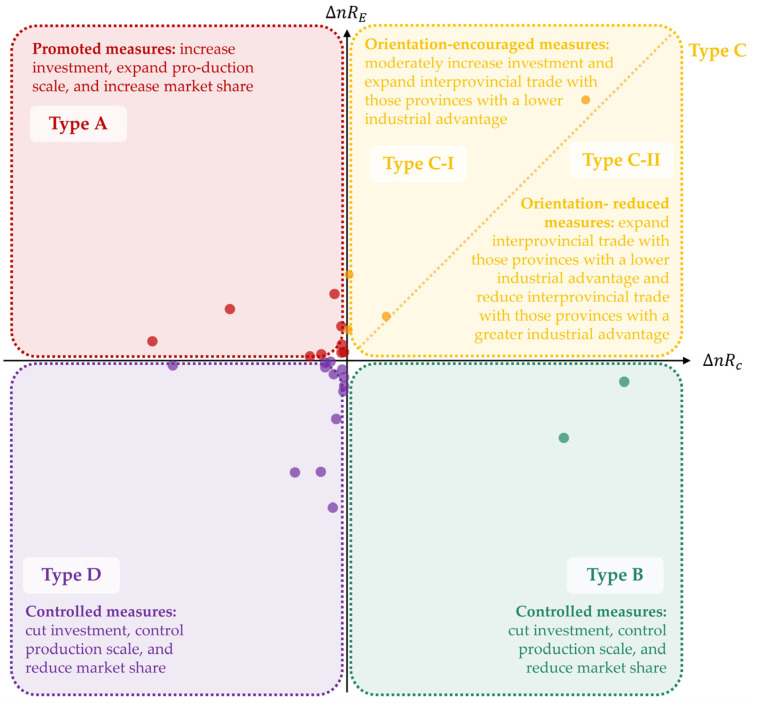
Improvement measures for each industrial type.

**Figure 10 ijerph-19-17033-f010:**
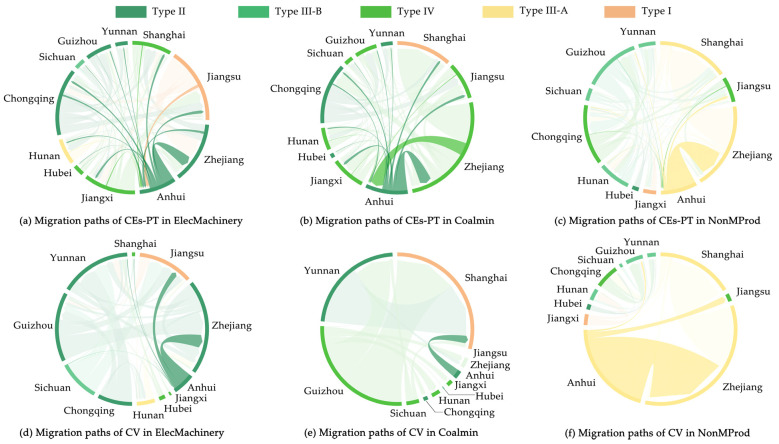
Migration paths of ECEs-IPT and CV in major industry sectors in Anhui among the YREB in 2017.

**Table 1 ijerph-19-17033-t001:** The basic structure of the MRIO table.

		Intermediate Use							Interprovincial Outflow from Region r				
region r	industry sector	$S_{1}$	⋯	$S_{j}$	⋯	$S_{n}$	final use	Total output	$R_{1}$	⋯	$R_{d}$	⋯	$R_{m}$
intermediate input	$S_{1}$	$x_{r}^{11}$	⋯	$x_{r}^{1j}$	⋯	$x_{r}^{1n}$	$F_{r}^{1}$	$X_{r}^{1}$	$PEX_{r1}^{1}$	⋯	$PEX_{rd}^{1}$	⋯	$PEX_{rm}^{1}$
	⋮	⋮		⋮		⋮	⋮	⋮	⋮		⋮		⋮
	$S_{i}$	$x_{r}^{i1}$	⋯	$x_{r}^{ij}$	⋯	$x_{r}^{in}$	$F_{r}^{i}$	$X_{r}^{i}$	$PEX_{r1}^{i}$	⋯	$PEX_{rd}^{i}$	⋯	$PEX_{rm}^{i}$
	⋮	⋮		⋮		⋮	⋮	⋮	⋮		⋮		⋮
	$S_{n}$	$x_{r}^{n1}$	⋯	$x_{r}^{nj}$	⋯	$x_{r}^{nn}$	$F_{r}^{n}$	$X_{r}^{n}$	$PEX_{r1}^{n}$	⋯	$PEX_{rd}^{n}$	⋯	$PEX_{rm}^{n}$
value-added		$VA_{r}^{1}$	⋯	$VA_{r}^{j}$.	⋯	$VA_{r}^{n}$							
total input		$Y_{r}^{1}$	⋯	$Y_{r}^{j}$	⋯	$Y_{r}^{n}$							
interprovincial inflow into region r	$R_{1}$	$PIM_{1r}^{1}$	⋯	$PIM_{1r}^{j}$	⋯	$PIM_{1r}^{n}$							
	⋮	⋮		⋮		⋮							
	$R_{d}$	$PIM_{dr}^{I}$	⋯	$PIM_{dr}^{j}$	⋯	$PIM_{dr}^{i}$							
	⋮	⋮		⋮		⋮							
	$R_{m}$	$PIM_{mr}^{1}$	⋯	$PIM_{mr}^{j}$	⋯	$PIM_{mr}^{n}$							

## Data Availability

Not applicable.

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
