# Peer review of "Spatial–Temporal Evolution and Improvement Measures of Embodied Carbon Emissions in Interprovincial Trade for Coal Energy Supply Bases: Case Study of Anhui, China"

_ijerph, 2022, doi:10.3390/ijerph192417033_

Round 1

Reviewer 1 Report

The authors' paper "Spatial-Temporal Evolution and Improvement Measures of 2 Embodied Carbon Emissions in Interprovincial Trade for Coal 3 Energy Supply Bases under Regional Economic Integration: 4 Case Study of Anhui, China" is written in simple and accessible language for readers. The study presented in the article is of high relevance. I believe that the authors are engaged in an important direction for the energy industry.

But despite the positive impression of the article, I have several recommendations and comments for the authors:

1.    Line 81, too many links. I suggest breaking it down into several. No more than three.

2.    Line 106, Target for the model is based on 2017 data. Not the best solution, as this given is not true. There was a covid-19 pandemic in 2019, a cold in Europe in 2020, etc. I recommend reconsidering the period of time at least until 2020, and preferably until 2021.

3.    Figure 1 is of very poor resolution. I recommend to improve.

4.    Figure 2 is also very poor quality. If the figure is borrowed, the authors must be indicated. The data is for 2017. I think they need improvement.

5.    Figure 3 data is for 2017. I think they need improvement.

6.    The data in Figure 4 are for 2017 and are of poor resolution. I think they need improvement.

7.    The whole model is based on outdated data, it is necessary to revise the model.

8.    Figure 5 is borrowed and of very poor resolution. Needs to be improved. There is no link to the authors, I recommend redrawing or linking.

9.    Figures 6,7,8,9,10 require serious revision. There are no references to the authors.

10. The conclusion is extensively written, needs to be improved for understanding.

The work is very interesting, but it needs to be reworked very seriously. Many of the pictures are not entirely clear. These are the data of the authors or taken from the literature. What is the novelty of the work of the authors. The work is rather weak and is based on the description of statistical data, while 5 years from 2017 to 2022 are not taken into account.

I see potential for further research in the work of the authors. I look forward to the revised article.

Reviewer 2 Report

After carefully reading the manuscript entitled: “Spatial-Temporal Evolution and Improvement Measures of Embodied Carbon Emissions in Interprovincial Trade for Coal Energy Supply Bases under Regional Economic Integration: Case Study of Anhui, China”, I noticed that it presents a high scientific contribution, so I recommend the ACCEPT of the manuscript for publication with minor corrections:

1 - I think this title is more appropriate: “Spatial-Temporal Evolution and Improvement Measures of Embodied Carbon Emissions in Interprovincial Trade for Coal Energy Supply Bases: Case Study of Anhui, China”.

2 - The Abstract is good, well founded and clear. Congratulations to the authors.

3 - In item: “2. Study Area Materials and Methods”, I suggest leaving only "2. Materials and Methods".

4 - Figure 5 does not show an adequate resolution, I believe this should be reviewed.

5 - Figure 6 and 7 does not have an adequate resolution, I believe this should be reviewed, and should be presented on a larger scale

6 - The English writing is of great quality.

7 - Congratulations to the authors.

Author Response

请参阅附件。

Round 2

Reviewer 1 Report

I thank the authors for the deep revision of the manuscript. Now it has become more understandable and interesting for future readers. I thank them for taking my recommendations seriously and correcting them to a greater extent. I believe that the work in this form can be published.